# SARS-CoV-2 Infection Susceptibility of Pregnant Patients at Term Regarding ABO and Rh Blood Groups: A Cohort Study

**DOI:** 10.3390/medicina57050499

**Published:** 2021-05-14

**Authors:** Roxana Covali, Demetra Socolov, Ioana Pavaleanu, Alexandru Carauleanu, Vasile Lucian Boiculese, Razvan Socolov

**Affiliations:** 1Department of Radiology, Elena Doamna Obstetrics and Gynecology University Hospital, 700398 Iasi, Romania; 2Department of Obstetrics and Gynecology, Cuza Voda Obstetrics and Gynecology University Hospital, 700038 Iasi, Romania; socolov@hotmail.com (D.S.); acarauleanu@yahoo.com (A.C.); 3Department of Obstetrics and Gynecology, Elena Doamna Obstetrics and Gynecology University Hospital, 700398 Iasi, Romania; ioana-m-pavaleanu@umfiasi.ro (I.P.); socolovr@yahoo.com (R.S.); 4Department of Statistics, Grigore T. Popa University of Medicine and Pharmacy, 700115 Iasi, Romania; lboiculese@gmail.com

**Keywords:** SARS-CoV-2, pregnant patients, ABO blood group, Rh blood group, infection risk

## Abstract

*Background and Objectives*: The susceptibility of pregnant patients at term to SARS-CoV-2 infection regarding the ABO and Rh blood group polymorphism was analyzed in this study. *Materials and Methods:* In this prospective study, 457 patients admitted for delivery at term in our hospital, between 1 April 2020 and 31 December 2020 were studied. There were 46 positive and 411 SARS-CoV-2 negative patients. Their values for RT-PCR, ABO, and Rh blood group analyses, which were determined upon admittance, were studied. *Results:* A slightly higher percentage of infected pregnant patients at term belonged to the A blood group compared with the percentage belonging to the other blood groups; this was also true for the healthy control group. For the Rh-negative pregnant patients at term, the odds of being infected with SARS-CoV-2 was OR = 1.22 compared with Rh-positive patients where OR = 1. In our study, the highest risk was found among BIII Rh-negative pregnant patients at term (OR = 3). None of the above differences were statistically significant. *Conclusions:* No significant difference was found between the percentage of ABO or Rh blood groups in SARS-CoV-2 positive patients when compared with SARS-CoV-2 negative patients (*p* = 0.562).

## 1. Introduction

Maintenance of the ABO polymorphism throughout human evolution might have had an adaptative value in past epidemics of other viruses with transmission characteristics similar to those of SARS, because it allows for modifications of immune behavior which limits the spread of epidemics [1].

In 1971, Robinson [2] discovered that the combined estimate for the risk of having an E. coli infection for patients with blood groups B or AB was 1.55 times that of patients with blood groups O or A, for Salmonella infection it was 2.31 times the risk because these microorganisms possess a B-like antigen [3]. Several blood group proteins in the membrane act as receptors for intraerythrocytic pathogens, such as malaria [4,5]. In Hong Kong, blood group O individuals appear to be more resistant to SARS compared with non-O individuals [6].

Barnkob [7] identified ABO blood type as a risk factor for SARS-CoV-2 infection, but not for hospitalization or death from COVID-19. The relative risk was 0.87 for acquiring COVID-19 in patients belonging to blood group O compared with patients with other blood groups. Saify [8] demonstrated that although group A had an increased risk of developing COVID-19, the association did not reach statistical significance, whereas in the analysis of the combined phenotypes, the A- blood group had a significantly increased risk of COVID-19. Rahim [9] reported that a significant association existed between blood types B and AB and susceptibility to COVID-19, whereas there was no association between blood types A and O with COVID-19, and that Rh-D positive blood types are less susceptible to COVID-19.

Ray [10] discovered that the O and Rh-negative blood groups may be associated with a slightly lower risk for SARS-CoV-2 infection and severe COVID-19 illness. Bhandari [11] showed that although type A blood seems to be slightly more prevalent with respect to B and AB types in hospitalized patients, strong confounders of age and sex dilute this significance, and Rh-negative patients appear to have a higher mortality rate; although this too is strongly confounded.

Ahmed [12] recommended that pregnant women with blood group A would require extra vigilance from clinicians and may warrant more personal protection to lower the risk of COVID-19 infection.

Because no unanimous opinion has been reached so far, the aim of this study was to establish the susceptibility for SARS-CoV-2 infection in pregnant women at term regarding the ABO and Rh blood groups.

## 2. Materials and Methods

In a prospective study, all patients admitted for delivery at term to Elena Doamna Obstetrics and Gynecology University Hospital in Iasi, Romania, between 1 April 2020 (when we were designated a COVID-19 support hospital) and 31 December 2020 were included. Inclusion criteria: patients who delivered at term in our hospital whose blood analysis before delivery were performed in our hospital. Exclusion criteria: patients who delivered elsewhere and were afterward admitted to our hospital; patients who delivered in our hospital as soon as they arrived, so that blood harvest for analysis before delivery could not be performed; patients who had the blood analysis performed in another hospital and were then rushed to our hospital for delivery (because we are a COVID-19 support hospital, and others are not) were excluded from the study. The remaining 457 patients were included in two groups: group 1, SARS-CoV-2 positive patients (*n* = 46) and group 2, SARS-CoV-2 negative patients (*n* = 411). Except for the patients who came from another hospital or a quarantine zone, with a SARS-CoV-2 positive RT-PCR (real-time polymerase chain reaction) test within the last 14 days, all the other patients were RT-PCR tested upon arrival (Appendix A), kept or delivered separately in an intermediate zone, and based on the RT-PCR result they were admitted to the specialized SARS-CoV-2 positive patients’ area or the SARS-CoV-2 negative patients’ area. Among other tests, ABO and Rh blood groups were determined upon admittance, with MAN-HEMATO Laboratory Equipment.

Patients’ age ranged between 17–38 years old in group 1 (positive), and between 15–45 years old in group 2 (negative). Mean age, gestation, and parity number (27.83, 2.28, and 1.87 in positive patients versus 26.76, 2.56, and 2.15 in negative patients) was not significantly different between the two groups (*p* = 0.156; 0.441 and 0.143, respectively), nor were the median values of the age, gestation, and parity number (28, 2, and 2 in positive patients versus 27, 2, and 2 in negative patients; Table 1).

Written informed consent was obtained from all patients. Research Ethics Committee Approval from the Elena Doamna University Hospital was obtained for this study (Number 4 / 2 April 2020).

Statistical analysis was performed with SPSS version 18 software (PASW Statistics for Windows, SPSS Inc.: Chicago, IL, USA). For descriptive measures, we computed the mean, standard deviation, median, and quartile 1 and 3 (for non-normal distributions), minimum and maximum limits. To compare the data, the non-parametric Mann–Whitney U test and the Pearson chi-square test were applied according to data distribution. To assess the effect of blood groups on the probability of SARS-CoV-2 infection, the odds ratio and confidence interval were computed (95% probability). The standard cutoff significance of *p* = 0.05 was used to decide on hypothesis conclusions.

## 3. Results

### 3.1. ABO Blood Group

The number and percentage of different blood groups in the SARS-CoV-2 positive and negative pregnant patients at term are shown in Table 2. Among the SARS-CoV-2 positive pregnant patients at term, most (52%) belonged to the AII blood group, and only 6.5% belonged to the ABIV blood group. The positive likelihood ratio was the highest (11.7%) in the AII blood group of pregnant patients at term, and the lowest (7.1%) was in the OI blood group of pregnant patients at term. The Pearson chi-square test showed no significant difference between the percentage of ABO blood groups in SARS-CoV-2 positive patients compared with the SARS-CoV-2 negative patients (*p* = *0*.562). No ABO blood group of at term pregnant patients was more affected than the others.

### 3.2. Rh Blood Group

The number and percentage of Rh blood group types in the SARS-CoV-2 positive and negative pregnant patients at term are shown in Table 3. In the SARS-CoV-2 positive pregnant patients at term, most (87%) belonged to the Rh-positive blood group, and only 13% belonged to the Rh-negative blood group. The positive likelihood ratio was higher (11.8%) in the Rh-negative blood group of pregnant patients at term, and lower (9.8%) in the Rh-positive blood group of pregnant patients at term. Based on the Pearson chi-square test, no significant difference was found between the percentage of Rh type in SARS-CoV-2 positive patients compared with the SARS-CoV-2 negative patients (*p* = 0.669). The Rh factor did not influence the possibility of being SARS-CoV-2 positive among pregnant patients at term.

### 3.3. ABO and Rh Blood Groups Combined

The number and percentage of ABO and Rh combined blood groups in the SARS-CoV-2 positive and negative pregnant patients at term is shown in Table 4 and Table 5. In the SARS-CoV-2 positive pregnant patients at term, most (47.82%) belonged to the AII Rh-positive blood group, and 0% belonged to the ABIV Rh-negative blood group. The positive likelihood ratio was the highest (23.1%) in the BIII Rh-negative blood group of pregnant patients at term, and the lowest (0%) in the ABIV Rh-negative blood group of pregnant patients at term.

Because *p* < 0.05 was considered significant, no statistically significant differences were found between numbers or percentages of ABO and Rh blood groups of the SARS-CoV-2 positive and SARS-CoV-2 negative groups of patients.

There was no statistically significant difference. The most at-risk group was BIII Rh-negative, where the OR of 3 is unfortunately insignificant. The OR is calculated through comparison with the Rh-positive group in the BIII blood group. In other words, in the BIII group, the absence of the Rh factor would be a risk for SARS-CoV-2 infection, but this was not statistically significant.

Overall (meaning that we do not consider the ABO blood group), Rh-negative pregnant patients at term (OR = 1.22), compared with Rh-positive patients (OR = 1), are at higher risk for infection with SARS-CoV-2, but this is not statistically significant.

For the ABIV group, it is not possible to calculate the OR because there is a cell with 0 patients in it.

## 4. Discussion

We found no significant differences between the ABO and Rh type blood groups in susceptibility to acquiring the SARS-CoV-2 infection among pregnant patients at term. Though there was a slightly higher percentage of pregnant patients at term belonging to the A blood group compared with the percentage of the other blood groups, this was also true for the healthy control group. There was no statistical significance between the percentages in the two groups (*p* = 0.562). This is in accordance with what Bhandari [11] demonstrated, that blood group A patients seem to be slightly more numerous in the infected group than in the control group, but other factors are also involved, resulting in no statistically significant differences between ABO blood group distributions between the SARS-CoV-2 infected patients and the non-infected ones.

The ABO frequency distribution in the study by Ahmed et al. [12] showed that among uninfected pregnant women in Leicester, UK, the O blood group dominated (42%), and the A blood group frequency was second (35%); this was the same in Birmingham, UK, with 40% of uninfected pregnant patients being in the O blood group and 39% being in the A blood group. The opposite was found among the SARS-CoV-2 infected patients, where the A group patients dominated: 35% of patients were group A and 25% of patients were group O in Leicester, similar to the frequency of 40% A group patients and 24% O group patients.

In our study in Romania, the A group dominated among all pregnant patients, with 44% of the healthy pregnant patients and 52% of the SARS-CoV-2 infected patients being in this group, whereas the frequency of O group pregnant patients came in second: 28.7% of the healthy pregnant patients and 19.6% of the SARS-CoV-2 infected pregnant patients were in this group.

This may be explained by a population characteristic. In the United Kingdom, O blood group persons dominate, representing 47% of the population; however, in Romania, the A group persons dominate, representing 43% of the population [13]. If the SARS-CoV-2 virus infects significantly more A blood group patients than patients of the other blood groups, this increment would be visible, especially in populations where persons having a blood group other than A generally dominate. Still, this explanation is not true for other countries. In the United States, of the over 1000 patients who tested positive, Latz [14] reported 34.2% blood type A and 45.5% blood type O patients, and in the United States O blood group persons dominate [13].

Rahim [9] reported that, in Pakistan, there was increased susceptibility among persons belonging to B and AB blood groups, and, in Pakistan, B blood group persons dominate (38%) over O and A blood group persons (29% and 23%, respectively) [13]. This also does not explain why the frequency of AB blood group infected patients increased. According to Bhandari [11], other factors, including population characteristics, seem to be strongly involved in the immune response, besides just the ABO blood group. Khalil [15] also questioned the role of the ABO blood group system in dictating the severity of this disease.

In populations where the O group distribution was higher than the other blood groups, results varied again: in the United States, Leaf [16] reported a higher-than-expected frequency of blood type A and a lower-than-expected frequency of blood type O among White patients, and no difference in the observed versus expected distribution of ABO phenotypes among Black or Hispanic patients. Another study in the United States, conducted by Latz [14], showed that blood type A had no correlation with positive testing, but blood type B and AB were associated with higher odds of testing positive for the disease. In China, Li [17] found that the proportion of blood group A among patients infected with SARS-CoV-2 was significantly higher than that among healthy controls, whereas the proportion of blood group O among SARS-CoV-2 infected patients was significantly lower than among healthy controls. In Canada, Ray [10] found that there was a lower risk for severe COVID-19 illness or death associated with the type O blood group versus not only the A group but all other groups. In Iran, Abdollahi [18] reported a higher rate of infection among patients in the AB histo-blood group, and patients in the O histo-blood group had a lower rate of infection.

In populations where the A blood group dominated, situation varied, too. In Romania, we report no significant difference in regards to ABO blood groups between positive and negative pregnant patients at term. In Denmark, Barnkob [7] reported that among the SARS-CoV-2 positive individuals, considerably fewer group O individuals were noted; when the O blood group was excluded, no significant differences were seen among A, B, and AB groups.

In populations where the B group distribution was higher than the other groups, situation also varied. In Pakistan, Rahim [9] reported a significant association between blood types B and AB and susceptibility to COVID-19. In Afghanistan, Saify [8] reported that being in the A^-^ blood group remarkably increased the risk of SARS-CoV-2.

As regards the Rh blood groups, Rahim [9] noticed that Rh-D positive blood types are less susceptible to COVID-19. On the contrary, Latz [14] reported that Rh-positive status was associated with higher odds of testing positive for the SARS-CoV-2 virus. Zietz [19] reported the Rh-negative blood type to have a protective effect for three outcomes: infection, intubation, and death. Ray [10] reported a lower risk for severe COVID-19 illness or death associated with the Rh-negative versus the Rh-positive blood group. Bhandari [11] reported no significant relationships between Rh blood types and susceptibility or mortality with COVID-19 infection in the United States. Abdollahi [18] noted that the Rh blood group phenotype was not statistically significant in determining a patient’s vulnerability. As regards the Rh blood groups, we found that the Rh factor did not significantly influence the possibility of being SARS-CoV-2 positive among pregnant patients at term. For the Rh-negative pregnant patients at term (OR = 1.22), compared with Rh-positive patients (OR = 1), the risk of being infected with SARS-CoV-2 was higher, but it was not statistically significant. In our study, the highest risk was found among BIII Rh-negative pregnant patients at term (OR = 3); in BIII blood group patients, the absence of Rh factor would be a risk factor for SARS-CoV-2 infection compared with BIII Rh-positive patients, but this was not statistically significant.

Though it would be very interesting to discuss the correlation between the severity of the disease in pregnant patients at term and the blood types, all of the SARS-CoV-2 positive pregnant patients at term admitted to our hospital last year (the interval we studied in this article) had the mild form of the disease. We only had one second trimester pregnant patient with a severe form of COVID-19. However, between January 1st and April 15th of 2021, several pregnant patients with severe and moderate cases of COVID-19 were admitted to our hospital (we had fewer positive patients, but there were more complicated cases of COVID-19, probably due to the spread of the British variant). Because only one patient was pregnant at term, and most of the others were in the second and third trimesters of pregnancy, we added some tables about second and third trimester pregnant patients’ ABO and Rh blood groups and the severity of COVID-19 disease in Appendix B.

## 5. Conclusions

We found no significant differences between the different ABO and Rh blood groups in susceptibility of acquiring a SARS-CoV-2 infection in pregnant patients at term. A slightly higher percentage of pregnant patients at term belonged to the A blood group compared with the percentage of the other blood groups; this was also true for the healthy control group. There was no statistically significant difference between the percentages in the two groups (*p* = 0.562). As regards the Rh blood groups, we found that the Rh factor did not significantly influence the possibility of being SARS-CoV-2 positive among pregnant patients at term. For the Rh-negative pregnant patients at term (OR = 1.22), compared with Rh-positive patients (OR = 1), the risk of being infected with SARS-CoV-2 was higher, but it was not statistically significant. In our study, the highest risk was found among BIII Rh-negative pregnant patients at term (OR = 3); in BIII blood group patients, the absence of Rh factor would be a risk factor for SARS-CoV-2 infection, compared with BIII Rh-positive patients, but this was not statistically significant.

## Figures and Tables

**Table 1 medicina-57-00499-t001:** Patient characteristics: mean values (and standard deviations) on the upper line, and median values (quartile 1, quartile 2) on the lower line of each value below.

Pregnant Patients at Term	SARS-CoV-2 Positive	SARS-CoV-2 Negative	Significance, *p*
Age (years)	27.83 (±5.28) 28 (23, 32)	26.76 (±6.27) 27 (22, 31)	0.156
Gestation (number)	2.28 (±1.3) 2 (1, 3)	2.56 (±1.86) 2 (1, 3)	0.441
Parity (number)	1.87 (±1.14) 2 (1, 3)	2.15 (±1.43) 2 (1, 3)	0.143

**Table 2 medicina-57-00499-t002:** Number and percentage of different blood groups in the SARS-CoV-2 positive and negative pregnant patients at term.

Pregnant Patients at Term Blood Group	SARS-CoV-2 Positive	SARS-CoV-2 Negative	LR+ ^1^
OI	9 (19.6%)	118 (28.7%)	7.1%
AII	24 (52.2%)	181 (44.0%)	11.7%
BIII	10 (21.7%)	80 (19.5%)	11.1%
ABIV	3 (6.5%)	32 (7.8%)	8.6%

^1^ LR+ represents the positive likelihood ratio.

**Table 3 medicina-57-00499-t003:** Number and percentage of Rh type in the SARS-CoV-2 positive and negative pregnant patients at term.

Pregnant Patients at Term Rh Factor	SARS-CoV-2 Positive	SARS-CoV-2 Negative	LR+ ^1^
Positive	40 (87.0%)	366 (89.1%)	9.8%
Negative	6 (13.0%)	45 (10.9%)	11.8%

^1^ LR+ represents the positive likelihood ratio.

**Table 4 medicina-57-00499-t004:** Number and percentage of ABO and Rh combined blood groups in the SARS-CoV-2 positive and SARS-CoV-2 negative pregnant patients at term, and significance *p*, according to the Fisher exact test.

Pregnant Patients at Term ABO and Rh Combined Blood Group	SARS-CoV-2 Positive	SARS-CoV-2 Negative	*p*	LR+ ^1^
OI Rh-positive	8 (17.39%)	104 (25.30%)	1.00	7.1%
OI Rh-negative	1 (2.17%)	14 (3.40%)		6.7%
AII Rh-positive	22 (47.82%)	163 (39.65%)	1.00	11.9%
AII Rh-negative	2 (4.34%)	18 (4.37%)		10%
BIII Rh-positive	7 (15.21%)	70 (17.03%)	0.155	9.1%
BIII Rh-negative	3 (6.52%)	10 (2.43%)		23.1%
ABIV Rh-positive	3 (6.52%)	29 (7.05%)	1.00	9.4%
ABIV Rh-negative	0 (0%)	3 (0.72%)		0%

^1^ LR+ represents the positive likelihood ratio.

**Table 5 medicina-57-00499-t005:** Risk estimate for the different ABO and Rh blood groups combined.

Blood Groups	OR	95% Confidence Interval	
		Lower	Upper
OI (Rh −/+)	0.929	0.108	7.991
AII (Rh −/+)	0.823	0.179	3.791
BIII (Rh −/+)	3.00	0.665	13.527
ABIV (Rh −/+)	-	-	-

OR = odds ratio. For the ABIV group, the OR could not be calculated because there were 0 patients in the Rh-negative SARS-CoV-2 positive patient group.

## Data Availability

All the data are available from the corresponding author upon reasonable request.

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
