# Peer review of "SARS-CoV-2 Infection Susceptibility of Pregnant Patients at Term Regarding ABO and Rh Blood Groups: A Cohort Study"

_medicina, 2021, doi:10.3390/medicina57050499_

Round 1

Reviewer 1 Report

In this manuscript, Covali R et al., have tried to examine the relationship between SARS-CoV-2 infection in pregnant patients and their blood types. Although many papers about COVID-19 has been published, there are few articles about this correlation, so people all over the world would have high expectations regarding the findings.

As a result of reviewing your manuscript, I have recommended some corrections, which I believe will improve the readability of the paper. The authors should correct the points listed below.

  1. In Table 1, the title is “Patient characteristics: mean values (and standard deviation)”. I think upper values (ex. 27.83±28) in each parameter would certainly be the mean values ± standard deviations as the title suggests, but what do the lower values indicate (ex. 28 (23, 32) )? You should explain it in the manuscript and give an appropriate title of Table 1. In addition, you have to add each unit, next to these parameters’ name (ex. Age → Age (years)).

  1. In Table 2-4, you should add positive ratio in each blood type group. For example, in table 2, the positive ration in OⅠblood group is 7.1% (≒9/(9+118)). It would be easier to compare each group if these numbers were provided.

  1. The authors only examined the relationship between blood types and SARS-CoV-2 in the presence (positive) or absence (negative) of the disease. However, is there any correlation between the severity of the disease and blood types? It would be very interesting to discover and discuss which blood type is more or less likely to be severely infected to COVID-19 in pregnant patients. Do you have any medical data about the severity of the disease? You had better compare the correlations between the conditions (mild, moderate, or severe) of COVID-19 and the patients’ blood types.

Author Response

Response to reviewer 1

Comments and Suggestions for Authors

In this manuscript, Covali R et al., have tried to examine the relationship between SARS-CoV-2 infection in pregnant patients and their blood types. Although many papers about COVID-19 has been published, there are few articles about this correlation, so people all over the world would have high expectations regarding the findings.

As a result of reviewing your manuscript, I have recommended some corrections, which I believe will improve the readability of the paper. The authors should correct the points listed below.

  1. In Table 1, the title is “Patient characteristics: mean values (and standard deviation)”. I think upper values (ex. 27.83±28) in each parameter would certainly be the mean values ± standard deviations as the title suggests, but what do the lower values indicate (ex. 28 (23, 32) )? You should explain it in the manuscript and give an appropriate title of Table 1. In addition, you have to add each unit, next to these parameters’ name (ex. Age → Age (years)).

Thank you for your kind suggestions. We changed as follows, HIGHLIGHTED IN YELLOW:

-what the lower values indicate (lines 98-99):

Table 1. Patient characteristics: mean values (and standard deviation) on the upper line, and median values (quartile 1, quartile 2) on the lower line of each value below.

-we also explained it in the manuscript above, lines 93-97:

Mean age, gestation and parity number (27.83; 2.28 and 1.87 in positive patients versus 26.76; 2.56 and 2.15 in negative patients) was not significantly different between the two groups (P= .156; .441 and .143, respectively), nor were the median values of the age, gestation and parity number (28; 2 and 2 in positive patients versus 27; 2 and 2 in negative patients). (Table 1).

and we completed the title, as shown above, lines 98-99:

Table 1. Patient characteristics: mean values (and standard deviation) on the upper line, and median values (quartile 1, quartile 2) on the lower line of each value below.

-we added each unit, lines 99-100, in Table 1:

Pregnant patients at term

SARS-CoV-2 positive

SARS-CoV-2 negative

Significance, P

Age (years)

27.83 (±5.28)

28 (23, 32)

26.76 (±6.27)

27 (22, 31)

.156

Gestation(number)

2.28 (±1.3)

2 (1, 3)

2.56 (±1.86)

2 (1, 3)

.441

Parity (number)

1.87 (±1.14)

2 (1, 3)

2.15 (±1.43)

2 (1, 3)

.143

  1. In Table 2-4, you should add positive ratio in each blood type group. For example, in table 2, the positive ration in OⅠblood group is 7.1% (≒9/(9+118)). It would be easier to compare each group if these numbers were provided.

Thank you for your kind suggestions. We changed as follows, HIGHLIGHTED IN YELLOW:

-Table 2, lines 126-128:

Table 2. Number and percentage of different blood groups in the SARS-CoV-2 positive and negative pregnant patients at term                       

Pregnant patients at term blood group

SARS-CoV-2 positive

SARS-CoV-2 negative

LR+ 1

OI

9 (19.6%)

118 (28.7%)

7.1%

AII

24 (52.2%)

181 (44.0%)

11.7%

BIII

10 (21.7%)

80 (19.5%)

11.1%

ABIV

3 (6.5%)

32 (7.8%)

8.6%

1LR+ represents the positive likelihood ratio.

-Table 3, lines 140-142

Table 3. Number and percentage of Rh type in the SARS-CoV-2 positive and negative pregnant patients at term                                                                                     

Pregnant patients at term

Rh factor

SARS-CoV-2 positive

SARS-CoV-2 negative

LR+ 1

Positive

40 (87.0%)

366 (89.1%)

9.8%

Negative

6 (13.0%)

45 (10.9%)

11.8%

1LR+ represents the positive likelihood ratio.

-Table 4, lines 153-156

Table 4. Number and percentage of ABO and Rh combined blood groups in the SARS-CoV-2 positive and in SARS-CoV-2 negative pregnant patients at term, and significance P, according to the Fisher exact test

Pregnant patients at term ABO and Rh combined blood group

SARS-CoV-2 positive

SARS-CoV-2 negative

P

LR+ 1

OI Rh positive

8 (17.39%)

104 (25.30%)

1.00

7.1%

OI Rh negative

1 (2.17%)

14 (3.40%)

6.7%

AII Rh positive

22 (47.82%)

163 (39.65%)

1.00

11.9%

AII Rh negative

2 (4.34%)

18 (4.37%)

10%

BIII Rh positive

7 (15.21%)

70 (17.03%)

.155

9.1%

BIII Rh negative

3 (6.52%)

10 (2.43%)

23.1%

ABIV Rh positive

3 (6.52%)

29 (7.05%)

1.00

9.4%

ABIV Rh negative

0 (0%)

3 (0.72%)

0%

 1LR+ represents the positive likelihood ratio.

  1. The authors only examined the relationship between blood types and SARS-CoV-2 in the presence (positive) or absence (negative) of the disease. However, is there any correlation between the severity of the disease and blood types? It would be very interesting to discover and discuss which blood type is more or less likely to be severely infected to COVID-19 in pregnant patients. Do you have any medical data about the severity of the disease? You had better compare the correlations between the conditions (mild, moderate, or severe) of COVID-19 and the patients’ blood types.

Thank you for your kind suggestions. We ADDED as follows, HIGHLIGHTED IN YELLOW:

-In Discusion, we added lines 257-267:

Though it would be very interesting to discuss the correlation between the severity of the disease in pregnant patients at term and the blood types, all the SARS-Cov-2 positive pregnant patients at term admitted to our hospital last year (the interval we studied in this article) had the mild form of disease. We only had one second trimester pregnant patient with a severe form of COVID-19. However, between January, 1st and April, 15th, 2021, several severe and moderate cases of COVID-19 in pregnant patients were admitted to our hospital (we had fewer positive patients, but more complicated cases of COVID-19, probably due to the spread of the British variant). Because only one patient was at term pregnant, and most of the others were second and third trimester pregnant, we added some tables about second and third trimester pregnant patients’ ABO and Rh blood groups and the severity of COVID-19 disease in Appendix B.

-we added Appendix B, lines 310-331:

Appendix B

  The correlation between ABO and Rh blood groups of the pregnant patients at term and the severity of COVID-19 disease could not be studied so far in our hospital, because all the Sars-Cov-2 positive pregnant patients at term admitted during the study period had the mild form of COVID-19. We only had one severe case of COVID-19 in a second trimester pregnant patient. However, between January 1st, 2021 and April, 15th, 2021, several moderate and severe cases of COVID-19 were admitted in our hospital. Because only one patient was at term pregnant, and most admitted patients were second and third trimester pregnant, we studied all the SARS-Cov-2 positive second and third trimester pregnant patients admitted to our hospital during this period. (Tables B1-B3):

Table B1. Mild, moderate and severe form of COVID-19 in second and third trimester pregnant patients, according to blood group

Pregnant patients’ ABO blood group

Mild

Moderate

Severe

OI

2 (28.5%)

2 (28.5%)

3 (42.8%)

AII

3 (100%)

0 (0%)

0 (0%)

BIII

0 (0%)

0 (0%)

0 (0%)

ABIV

1 (50%)

0 (0%)

1 (50%)

Table B2. Mild, moderate and severe form of COVID-19 in second and third trimester pregnant patients, according to Rh blood group

Pregnant patients’

Rh blood group

Mild

Moderate

Severe

Positive

4 (40%)

2 (20%)

 4 (40%)

Negative

2 (100%)

0 (0%)

0 (0%)

Table B3. Mild, moderate and severe form of COVID-19 in second and third trimester pregnant patients, according to ABO and Rh combined blood group

Pregnant patients’ ABO and Rh combined blood group

Mild

Moderate

Severe

OI Rh positive

1 (16.6%)

2 (33.3%)

3 (50%)

OI Rh negative

1 (100%)

0 (0%)

 0 (0%)

AII Rh positive

2 (100%)

0 (0%)

 0 (0%)

AII Rh negative

1 (100%)

0 (0%)

0 (0%)

BIII Rh positive

0 (0%)

0 (0%)

0 (0%)

BIII Rh negative

0 (0%)

0 (0%)

0 (0%)

ABIV Rh positive

1 (50%)

0 (0%)

1 (50%)

ABIV Rh negative

0 (0%)

0 (0%)

0 (0%)

Thank you very much for your kind  suggestions!  

Reviewer 2 Report

The manuscript "SARS-CoV-2 infection susceptibility of pregnant patients at term regarding ABO and Rh blood groups: a cohort study" is not original because previous manuscripts have analysed the same correlation, but it is  well structured. The number of patient is not so high, but considering the period in which it was performed it is acceptable as number of pregnant women. The statistical analyses is well conducted and the language is acceptable. It needs to be improved in the materials and methods because it looks confusing, particularly in the tables, where there is a distinction between the groups I, II, II and IV, but it is not explained in the text. It represents an interesting work and it gives the opportunity to give some more data on correlation between SARS-CoV-2 infection susceptibility and ABO and Rh blood groups in this pandemic period. 

Round 2

Reviewer 1 Report

The revised paper is well-written, and the authors have clearly worked hard to produce a comprehensive dataset and detailed description. This study represents a massive effort and it deserves to be published, as these data will most likely continue to be a unique resource well into the future.